# Reduced Oxidative Stress and Enhanced FGF21 Formation in Livers of Endurance-Exercised Rats with Diet-Induced NASH

**DOI:** 10.3390/nu11112709

**Published:** 2019-11-08

**Authors:** Janin Henkel, Katja Buchheim-Dieckow, José P. Castro, Thomas Laeger, Kristina Wardelmann, André Kleinridders, Korinna Jöhrens, Gerhard P. Püschel

**Affiliations:** 1Department of Nutritional Biochemistry, Institute of Nutritional Science, University of Potsdam, D-14558 Nuthetal, Germany; kdieckow@uni-potsdam.de (K.B.-D.); gpuesche@uni-potsdam.de (G.P.P.); 2Department of Medicine, Division of Genetics, Brigham and Women’s Hospital and Harvard Medical School, Boston, MA 02115, USA; jquintaaraujocastro@bwh.harvard.edu; 3Department of Molecular Toxicology, German Institute of Human Nutrition, D-14558 Nuthetal, Germany; 4Department of Experimental Diabetology, German Institute of Human Nutrition, D-14558 Nuthetal, Germany; Thomas.Laeger@dife.de; 5German Center for Diabetes Research (DZD), D-85764 München-Neuherberg, Germany; Kristina.Wardelmann@dife.de (K.W.); Andre.Kleinridders@dife.de (A.K.); 6Junior Research Group Central Regulation of Metabolism, German Institute of Human Nutrition, D-14558 Nuthetal, Germany; 7Institute of Pathology, Carl Gustav Carus University Hospital Dresden, D-01307 Dresden, Germany; korinna.joehrens@uniklinikum-dresden.de

**Keywords:** NAFLD, NASH, endurance exercise, FGF21, glucose intolerance, cholesterol, oxidative stress

## Abstract

Non-alcoholic fatty liver diseases (NAFLD) including the severe form with steatohepatitis (NASH) are highly prevalent ailments to which no approved pharmacological treatment exists. Dietary intervention aiming at 10% weight reduction is efficient but fails due to low compliance. Increase in physical activity is an alternative that improved NAFLD even in the absence of weight reduction. The underlying mechanisms are unclear and cannot be studied in humans. Here, a rat NAFLD model was developed that reproduces many facets of the diet-induced NAFLD in humans. The impact of endurance exercise was studied in this model. Male Wistar rats received control chow or a NASH-inducing diet rich in fat, cholesterol, and fructose. Both diet groups were subdivided into a sedentary and an endurance exercise group. Animals receiving the NASH-inducing diet gained more body weight, got glucose intolerant and developed a liver pathology with steatosis, hepatocyte hypertrophy, inflammation and fibrosis typical of NAFLD or NASH. Contrary to expectations, endurance exercise did not improve the NASH activity score and even enhanced hepatic inflammation. However, endurance exercise attenuated the hepatic cholesterol overload and the ensuing severe oxidative stress. In addition, exercise improved glucose tolerance possibly in part by induction of hepatic FGF21 production.

## 1. Introduction

The easy access to inexpensive, highly palatable energy-dense food is not only the cause for the current overweight and obesity pandemic but also is reason for the worldwide increase in the prevalence of non-alcoholic fatty liver disease (NAFLD) that currently affects about one quarter of the world population [1]. NAFLD is even more frequent in type-2 diabetic patients. More than every second type-2-diabetic patient suffers from NAFLD [2]. While transient storage of triglycerides is a physiological function of the liver, chronic overfeeding results in a pathological fat accumulation. Non-alcoholic fatty liver (NAFL) is defined as a condition in which over 5% of the hepatocytes are affected by macroscopic steatosis or in which liver triglyceride content exceeds 5.5% of the liver wet weight in absence of significant alcohol consumption ([3] and citations therein). In up to one-third of the patients with NAFLD, steatosis is accompanied by inflammation and fibrosis in non-alcoholic steatohepatitis (NASH) that ultimately may lead to terminal organ failure. Although early stages of NAFLD can be cured by a life-style intervention that eliminates its cause by food restriction and a weight reduction of more than 10%, many patients do not comply with this treatment and fail to achieve the treatment goal [4]. Even though several clinical trials are under way, currently no pharmacological treatment for NAFLD is approved [5]. In addition to food restriction, increase in physical activity and regular exercise have repeatedly been shown to improve hepatic steatosis even in the absence of a significant reduction of body weight. Physical exercise is hence considered a treatment option for NAFLD/NASH [6]. The beneficial impact of training on NASH progression can in part be attributed to the restoration of insulin sensitivity resulting in an improved suppression of lipolysis in adipose tissue and a reduced flux of fatty acids to the liver. The fatty acid burden of the liver is further reduced by redistribution of fatty acids towards β-oxidation in the working skeletal muscle and by the improved glucose utilization of insulin-sensitive peripheral organs reducing the amount of substrate available for de novo fatty acid synthesis in the liver. However, some studies indicate that liver injury in NAFLD patients can be reduced without major changes of the hepatic lipid accumulation [7,8], indicating that other mechanisms might exist to reduce inflammation and fibrosis. These mechanisms cannot be analyzed in humans but can only be studied in animal models. However, many animal models of NASH that are currently employed fail to reproduce the full spectrum of symptoms of human NASH that develops based on overweight and the metabolic syndrome [9,10,11] and hence are not suitable to study the intervention. In the current study, we established a rat model of diet-induced NASH with systemic symptoms similar to the human metabolic syndrome in which the impact of physical activity on NASH development can be analyzed. The data indicate that exercise attenuated the diet-induced hepatic cholesterol overload and the ensuing oxidative stress but nevertheless enhanced signs of inflammation. Exercise improved whole-body glucose tolerance to which an exercise-induced hepatic fibroblast growth factor 21 (FGF21) production might have contributed.

## 2. Materials and Methods

All chemicals were of analytical or higher grade and obtained from local providers unless otherwise stated.

Animals and experimental design. Male Wistar rats (RjHan:WI, Janvier, Le Genest-Saint-Isle, France) were housed in type III-cages at 21 ± 1 °C with a 12 h light/dark-cycle. 9-week old rats were randomly assigned to one of the following diet groups with free access to food and liquid for 7 weeks: standard chow diet (V153 R/M-H; Ssniff, Soest, Germany) (STD; 3.06 kcal/g with 65% of energy from carbohydrates, 25% of energy from protein, 10% of energy from fat, and drinking water) or NASH-inducing diet (Altromin, Lage, Germany) (NASH-diet; 4.50 kcal/g with 35% of energy from carbohydrates, 16% of energy from protein, 49% of energy from fat (soybean oil), 1.25% cholesterol, 0,5% sodium cholate, and 15% fructose in drinking water).

Each diet group was subdivided into a sedentary (sed) and an exercise (run) group with 8 animals per group. Rats were subjected to a treadmill running protocol five times per week essentially as previously described [12]. During a pre-training period 2 weeks prior the intervention rats were adapted to the treadmill (Exer 3/6, Columbus Instruments, Columbus, USA) with progressive running from 8 to 15 m/min and up to 30 min. Within the training period rats exercised with a progressively increased load from 30 min to 1 h every week and from 10 to 19 m/min weekly (See Appendix A for methodologic details).

Rats had access to wooden gnawing sticks to avoid excessive teeth growth. Body weight was measured weekly. Rats were killed 48 h after the last training session by cardiopuncture after pentobarbital anesthesia. Plasma and organs were snap-frozen in liquid nitrogen and stored at −70 °C for biochemical analysis, aliquots of the organs were fixed for histological examination. Animal experiments were performed according to the ARRIVE guidelines. Treatment of the animals followed the German animal protection laws and was performed with approval of the state animal welfare committee (LUGV Brandenburg, V3 2347).

In vivo experiments. Intraperitoneal glucose tolerance test was performed after 6 weeks feeding intervention after an overnight fast by i.p. injection of 20% glucose solution (B.Braun, Melsungen, Germany) (2 g/kg body weight). Glucose and insulin levels were measured at the times indicated by a glucose sensor (Breeze2, Bayer; Berlin, Germany) or an insulin ELISA kit (Alpco, Samel, NH, USA).

Plasma and tissue analysis. FGF21 concentrations in plasma were determined by a commercial ELISA (RD291108200R, BioVendor, Karasek, Czech Republic). Triglycerides and cholesterol in plasma and liver homogenates were determined by TRIGS-assay (Randox; Crumlin, UK) and cholesterol liquicolor (HUMAN, Wiesbaden, Germany), respectively. Further plasma parameters (NEFAs, ALAT and ASAT) were quantified by an automated analyzer (Cobas Mira S, Hoffmann-La Roche, Basel, Switzerland) with the appropriate commercially available reagent kits. Liver malondialdehyde concentrations were determined by a commercial TBARS assay kit (Cayman Chemicals, Ann Arbor, MI, USA).

Histology. Formalin-fixed and paraffin-embedded liver sections (2–3 µm) were stained with Hematoxylin & Eosin or Sirius Red (both Sigma-Aldrich, Taufkirchen, Germany). Histological steatosis, inflammation and fibrosis were graded according to the NASH activity score (NAS) [13,14] by a liver pathologist (KJ) blinded to the type of intervention. Quantification of histological staining of Sirius Red was performed by using ImageJ software (version ImageJ 1.51j8, Wayne Rasband, National Institutes of Health, Bethesda, MD, USA) in images of 25 randomly chosen fields of each liver as previously described [15].

Real-time RT-PCR analysis. RNA isolation, reverse transcription and qPCR were performed as previously described [16]. For the analysis of mitochondrial copy number tissue was homogenized in DNA lysis buffer (100 mM Tris-HCl, 5 mM EDTA, 0.2% SDS, 200 mM sodium chloride, pH 8.0, 1 mg/mL Proteinase K), incubated for 2 h at 50 °C and subsequently 10 min at 95 °C. After centrifugation the DNA in the supernatant was purified by ethanol precipitation and then used for qPCR analysis. Oligonucleotides were listed in Appendix A. Results are expressed as relative gene expression normalized to expression levels of reference genes (Hprt, Eef2 and Srsf4) according to the formula: fold induction = 2^(a−b) gene of interest^/2^(a−b) reference genes^. Parameter “a” is the arithmetic mean of all Ct-values from samples of the STD group and parameter “b” is the Ct-value of every single sample. For calculations with more than one reference gene the mean of the difference (a−b) of each reference gene was used.

Western blot and oxyblot analysis. Western blot was performed as described previously [17] with oxidative phosphorylation cocktail for Western blot (Abcam, Cambridge, UK), anti-SOD2, anti-IRS2, anti-IRS1, anti-phospho-Ser_307/612/632_-IRS1 antibodies (Cell Signaling Technology, Frankfurt a.M., Germany) and anti-Fasn antibody (Santa Cruz Biotechnology, Heidelberg, Germany) as well as Ponceau S or FastGreen-staining (Sigma-Aldrich, Taufkirchen, Germany) as a loading control. Oxyblot analysis was done as described [18] with anti-DNP antibody (Sigma-Aldrich, Taufkirchen, Germany). Visualization of immune complexes was performed by using chemoluminescence reagent in ChemiDoc™ Imaging System with ImageLab software (Bio-Rad, Munich, Germany).

Glutathion peroxidase activity assay. 10 mg rat liver were homogenized with bullet beads in Assay Buffer (100 mM Tris, 300 mM KCl, 0.1% Triton-X-100). After centrifugation (15 min, 4 °C, 10,000× *g*) homogenates were used for protein determination and glutathione peroxidase activity assay as described previously [19].

Statistical analysis. The statistical significance of differences was determined by two-way-ANOVA with Tukey’s post hoc test for multiple comparisons, Kruskal-Wallis non-parametric test with Dunn’s post hoc test for multiple comparisons or Student’s *t*-test for unpaired samples using GraphPad Prism version 8 for Windows (GraphPad Software, La Jolla, CA, USA). Differences with a *p* ≤ 0.05 were considered statistically significant.

## 3. Results

### 3.1. Exercise-Dependent Attenuation of Diet-Induced Glucose Intolerance

Male Wistar rats were fed either a standard chow diet (standard-diet) or a NASH-inducing high-fat, high-cholesterol, high-fructose diet (NASH-diet). Both diet groups were subdivided into a sedentary group (sed) and a group subjected to endurance exercise training (run). Both NASH-diet groups gained significantly more weight (NASH-diet sed, 211 ± 13 g; NASH-diet run, 190 ± 9 g; standard-diet sed, 145 ± 9 g; standard-diet run, 121 ± 9 g) during the intervention period of seven weeks than the control diet groups (Figure 1). Endurance exercise had no significant impact on weight gain in either diet group.

As expected, the consumption of the NASH-inducing diet went along with the development of glucose intolerance. When animals were subjected to an i.p. glucose tolerance test, blood glucose levels were higher and the area under the glucose curve was 20% greater in the sedentary NASH-diet group than in the corresponding standard-diet group (Figure 2A,B). Although endurance exercise did not impact diet-induced weight gain, it completely abolished the diet-induced glucose intolerance. As a further sign of diet-induced insulin resistance, the plasma insulin levels in the sedentary NASH-diet group were significantly higher than in the corresponding sedentary control diet group during the glucose tolerance test (Figure 2C). By contrast, plasma insulin concentrations in the NASH-diet endurance exercise group were indistinguishable from the control group. HOMA-IR as a parameter of insulin resistance was significantly two-fold elevated in sedentary rats fed a NASH-inducing diet compared to the sedentary control diet group while HOMA-IR was not different from control in rats fed the NASH-diet performing exercise training (Figure 2D). Thus, apparently endurance exercise improved insulin sensitivity without reducing body weight. Since it was reported that exercise might improve NASH independently of a weight reduction [20], it was assumed that the training intervention might have attenuated the diet-induced NASH development and the ensuing hepatic insulin resistance.

### 3.2. Exercise-Dependent Increase in Diet-Induced Hepatic Inflammatory Response

As expected, animals receiving the NASH-inducing diet developed liver pathology typical of NAFLD/NASH. Histological analysis of H & E-stained liver slices revealed both macro- and micro-vesicular steatosis, hypertrophy of hepatocytes and infiltration with inflammatory cells (Figure 3A). The NASH activity score (NAS) was determined by a liver pathologist blinded to the treatment group according to a scoring model adapted to rodents [14]. While none of the animals receiving the control diet displayed signs of NAFLD or NASH, all animals receiving the NASH-inducing diet had a pathological NASH activity score (Figure 3B): The NAS of five of the eight animals in the sedentary NASH-diet group had a NAS indicative of a NAFLD while the remaining three displayed clear signs of a NASH (with a total score above 4 of which at least one point was given for inflammation). Endurance exercise had no significant impact on the NAS. Three of seven animals in the NASH-diet training group were diagnosed with NAFLD while the remaining four had signs of NASH. Thus, at odds with expectations, endurance exercise did not improve the NAS.

This result was further corroborated by biochemical analysis of liver samples. The hepatic triglyceride and cholesterol content was significantly increased about 6-fold and 15-fold, respectively, in animals receiving the NASH-diet in comparison to the control diet (Figure 3C,D). While endurance exercise did not affect triglyceride accumulation it significantly reduced cholesterol accumulation by about 25%. The expression of fatty acid synthase as a measure of de novo lipogenesis was reduced both on the mRNA and on the protein level by the NASH-inducing diet but exercise did not impact its expression (Appendix A). Fibrosis was quantified by colorimetric analysis of the parenchymal areas of Sirius Red-stained liver slices. The Sirius Red staining intensity in parenchymal areas of livers of animals receiving the NASH-inducing diet was significantly about 13-fold higher than in livers of animals receiving control diet, however, no impact of training could be observed (Figure 3A,E).

As a biochemical marker of the macrophage activation and infiltration, the expression of the gene Adgre1 (F4/80) was determined in liver samples. As expected the expression was significantly increased about 1.7-fold in sedentary animals receiving the NASH-diet (Figure 4A). Contrary to our expectations, endurance exercise did not attenuate F4/80 expression as a measure of macrophage infiltration and activation but rather enhanced it to 2.8-fold in comparison to sedentary standard-diet-fed animals. Similarly, the expression of the pro-inflammatory cytokine Interleukin-1β (IL-1β) was increased in livers of sedentary animals receiving the NASH-diet 1.8-fold (Figure 4B). As with F4/80, IL-1β expression was further enhanced to 3.3-fold by endurance exercise and not attenuated, as expected. In summary, endurance exercise appeared to enhance the diet-induced hepatic inflammation but to reduce the diet-induced hepatic cholesterol overload.

### 3.3. Exercise-Dependent Reduction of Diet-Induced Hepatic Cholesterol Overload and Oxidative Stress

The NASH-inducing diet has previously been shown to cause severe oxidative stress in the liver of mice [18] resulting in the formation of protein carbonyls. The hepatic cholesterol overload has been implicated in the development of this oxidative stress. In keeping with this observation in the current study, protein carbonyl formation was strongly enhanced in livers of both groups receiving the NASH-inducing diet. Most notably, the training intervention significantly reduced protein carbonyl formation (Figure 5A,B). Densitometric analysis of the band intensities of protein carbonyls in Western blots revealed a significant 40% reduction of protein carbonyls in the exercising NASH-diet groups in comparison to the sedentary NASH-diet group. Furthermore, formation of malondialdehyde as a parameter for lipid oxidation was significantly increased in livers of sedentary rats fed a NASH-diet compared to animals fed a standard diet. This increase was completely blunted in exercised rats fed a NASH-inducing diet (Figure 5C). Hepatic oxidative stress can be a consequence of diet-induced mitochondrial dysfunction [16]. In accordance with these previous findings, the expression of the complexes I and II of the respiratory chain were reduced in livers of NASH-diet-fed rats. While the training intervention improved complex I expression slightly but not significantly, it had no impact on complex II expression (Figure 5D, quantification in Appendix A).

Oxidative stress fosters the antioxidant defense system. As expected, protein expression of superoxide dismutase 2 (SOD2), a key anti-oxidative enzyme, was significantly enhanced by about 50% in livers of sedentary rats fed a NASH-diet compared to the liver of animals in the sedentary standard-diet group (Figure 5E,F). The expression of superoxide dismutase 2 was further augmented by the training in livers of exercised rats fed a NASH-diet. Similarly, the activity of hepatic glutathione peroxidases was improved in NASH-diet-fed rats by the training intervention (NASH-diet sed, 581 ± 137 mU/mg protein; NASH-diet run, 969 ± 122 mU/mg protein; *p* = 0.0566).

Thus, training intervention significantly reduced hepatic cholesterol overload as well as oxidative stress and tended to improve the anti-oxidative defense capacity. This reduction of oxidative stress in the liver might have alleviated hepatic insulin resistance without affecting the NAS.

Therefore, parameters indicative of hepatic insulin sensitivity were analyzed. On the mRNA level, insulin receptor substrate 2 (IRS2) expression was suppressed by the NASH-inducing diet (Appendix A). At variance with expectations, endurance exercise did not improve the expression of IRS2 but rather further reduced the IRS2 expression. These data were confirmed on the protein level, there was a tendency for a reduction of IRS2 by diet or exercise alone. However, only the exercising animals receiving the NASH-inducing diet showed a significant reduction of IRS2 in liver on the protein level (Appendix A). Since no adequate commercial antibodies are available, IRS2 phosphorylation could not be analyzed. Neither diet nor the exercise intervention caused a significant change in the inhibitory Ser_307,612/632_-phosphorylation of IRS1 in the liver (Appendix A).

### 3.4. Possible Contribution of Exercise-Dependent Hepatic FGF21 Production to the Reduction of Diet-Induced Systemic Glucose Intolerance

Thus, endurance exercise contrary to expectations did not improve hepatic steatosis or fibrosis and appeared to enhance rather than to attenuate hepatic inflammation in animals receiving a NASH-inducing diet. Although there was a slight reduction of the signs of diet-induced severe oxidative stress by exercise, it was unlikely that an exercise-induced improvement of hepatic insulin sensitivity contributed to the improved glucose tolerance. However, the liver still might have contributed indirectly to the improvement of systemic glucose tolerance. One possible candidate that might mediate such an effect is the hepatokine fibroblast growth factor 21 (FGF21) [21].

The hepatic expression of FGF21 has previously been shown to be under the control of PPARα and ChREBP that can be activated by fatty acids and the pentose phosphate cycle intermediate xylulose-5-phosphate ([22] and references therein). As expected, the hepatic expression of FGF21 was therefore induced about 5-fold in animals receiving the high-fat, high-fructose NASH-inducing diet. Notably, this induction was further increased by endurance exercise resulting in an almost 13-fold induction in these animals (Figure 6A). Hepatic FGF21 expression was mirrored by a corresponding increase in circulating FGF21 levels (Figure 6B). In contrast to liver, FGF21 expression in skeletal muscle was affected neither by the diet nor by exercise (Appendix A). Thus, an exercise-dependent induction of hepatic FGF21 expression increasing the plasma concentration of FGF21 might have contributed to improve systemic glucose tolerance.

Despite a low expression of the essential co-receptor β-klotho, muscle cells have been shown to be direct targets of FGF21 which increased glucose uptake in myotubes by the induction of Glut1 and protected cardiomyocytes from oxidative stress by the induction of uncoupling proteins [23,24]. Soleus muscle of rats was analyzed for potential effects of FGF21. Neither NASH-diet nor endurance exercise-induced Glut1 in the soleus muscle (Appendix A). Among several potential FGF21 target genes tested, only the gene for the mitochondrial uncoupling protein 3 (UCP3) was significantly induced about 3-fold in soleus muscle of trained rats receiving the NASH-inducing diet (Figure 7A). Diet itself caused a slight but non-significant increase in UCP3. 

The relative copy number of mitochondrial DNA in comparison to genomic DNA can be used as a biochemical marker for mitochondrial number. mtDNA copy numbers were significantly higher in soleus muscle of both in trained animals receiving standard diet and in sedentary animals receiving the NASH-inducing diet than in control animals (both about 1.4-fold). Most notably, training further enhanced the diet-dependent increase in mtDNA copy number in animals receiving the NASH-inducing diet (two-fold over sedentary standard-diet control) (Figure 7B).

## 4. Discussion

NAFLD is considered the hepatic manifestation of the metabolic syndrome and apart from the liver pathology is thought to contribute to the systemic symptoms of the metabolic syndrome, in particular dyslipidemia and the glucose intolerance. Since currently there is no accepted pharmacological treatment for NAFLD, life-style interventions are the only treatment options. Reduction of food intake with the goal of at least 10% weight reduction has been shown to better both hepatic pathology and systemic symptoms of the metabolic syndrome. However, compliance with such an intervention is poor. Physical exercise, on the other hand, has been shown to improve hepatic pathology in NAFLD patients even in the absence of a marked reduction of body weight [7]. The current study aimed to reproduce these results in an animal experiment to potentially allow mechanistic analysis.

### 4.1. Exercise-Dependent Augmentation of Hepatic Inflammation

Contrary to expectations NASH-diet-induced liver pathology was not improved and NAS was not affected by endurance exercise in rats (Figure 3). At odds with the expectation, the hepatic expression of IL-1β and F4/80 were even increased by endurance exercise (Figure 4). Acute treadmill running has been shown to induce IL-1β in the liver of mice [25]. However, to exclude the impact of acute exercise animals were rested two days before sacrifice in the current study. Similarly, overtraining has been shown to trigger inflammation in several organs and to cause a systemic increase in pro-inflammatory cytokines [26,27]. However, overtraining is not likely to be the cause of the pro-inflammatory response in the current study, because (1) the training protocol employed has previously been shown to attain less than 75% VO_2max_ in rats [28] and to reduce cancer cachexia-induced inflammation in adipose tissue. (2) According to the regulations of the animal welfare committee running speed had to be adjusted to avoid repeated contact of the animals with the shocker grid of the treadmill. Therefore, it is extremely unlikely that rats were running at the exhaustion limit that is necessary to induce chronic inflammation by overtraining.

Most strikingly, in a similar study [29] the authors described a reduction of a NASH-diet-induced hepatic IL-1β expression by endurance exercise. Apart from minor differences in the diet, a major difference between this and the current study is the rat strain used. While Wistar rats were used in the current study, Sprague Dawley rats were used in the previously published study. To test the hypothesis that different rat strains might react differentially to endurance exercise intervention, the current experiment was repeated with a small group of 16 Sprague Dawley animals (four per group). The results obtained were largely identical with the results obtained with the Wistar rats (Appendix A) with the notable exception of hepatic IL-1β expression: While in Sprague Dawley rats the diet-dependent induction of IL-1β was larger than that observed in Wistar rats, training tended to reduce the expression (Appendix A). Thus, possibly the impact of an endurance exercise training intervention on diet-induced NASH might differ between different rat strains. However, for a final proof of such a hypothesis, a larger number of animals of both strains would have to be tested in parallel under otherwise completely identical conditions.

### 4.2. Impact of Exercise on Severe Oxidative Stress in Liver

A high-fat, high-cholesterol NASH-inducing diet has recently been shown to cause severe oxidative stress in the liver [18]. Comparison of different diets revealed that most likely the combination of dietary cholesterol and ω6-poly-unsaturated fatty acids is mandatory to induce this severe oxidative stress [16]. Notably, in the current study endurance exercise only improved two parameters of the diet-induced liver pathology: The accumulation of cholesterol (Figure 3D) and the formation of protein carbonyls and malondialdehyde as parameters for mild and severe oxidative stress (Figure 5A–C). Thus, it is possible that the reduced hepatic cholesterol accumulation is causally related to the reduction in oxidative stress. In accordance with such a hypothesis, a cholesterol overload of the liver has been shown to cause severe oxidative stress in a bile duct ligation model of liver injury [30]. Similarly, the formation of specific oxysterols has been observed in a lipidomic study [31] in mice receiving a NASH-inducing diet. The oxysterols have been implicated in the progression to NASH. Progression to NASH has been associated with a depletion of antioxidants and mitochondrial damage that may be caused by oxysterols [32] and reduction of oxidative stress has been discussed to be one of the mechanisms by which physical exercise can improve NASH progression [33]. Oxidative stress may be an early event in the progression to NASH. Therefore, the seven-week duration of the intervention in the current study might not have been long enough to reveal an impact of exercise on NASH progression. In keeping with this interpretation, less than 50% of the animals in the current study had a NASH activity score indicative of NASH while over 90% of all mice receiving a comparable NASH-diet for 20 weeks had histologically confirmed signs of NASH [18].

### 4.3. Impact of Exercise on Glucose Tolerance

Although insulin signaling in the liver could not be tested, because animals were sacrificed after an overnight fast and no insulin was injected to activate the insulin signaling chain, several lines of evidence argue against an exercise-dependent improvement of hepatic insulin sensitivity. Most prominently, the major intracellular relay protein of the insulin receptor in the liver, the insulin receptor substrate, was down-regulated both on the mRNA and the protein level. Exercise enhanced rather than attenuated this down-regulation. The most reliable insulin target gene in rat liver is glucokinase. While the NASH-diet slightly induced glucokinase (Appendix A), this induction was attenuated by endurance exercise. Similarly, there was a diet-induced reduction in the expression of the key enzyme of gluconeogenesis, phosphoenolpyruvate carboxykinase (Pepck) (Appendix A). At variance of what would have been expected if exercise enhanced hepatic insulin sensitivity, exercise did not affect diet-dependent repression of Pepck. Thus, although the whole-body glucose tolerance was increased by endurance exercise, this was most likely not due to an improvement of the hepatic insulin sensitivity.

The second major sink for dietary glucose is skeletal muscle. Thus, training-dependent improvement of insulin sensitivity in skeletal muscle is a likely contributor to the improvement in systemic glucose tolerance. While direct training effects on skeletal muscle are likely important contributors, indirect effects might also contribute, among others the training-dependent production and release of hepatokines [34,35]. In this context, the observed exercise-dependent increase in FGF21 is of interest (Figure 6). FGF21 has been shown to improve glucose utilization by skeletal muscle in insulin-resistant mice [36]. This was assumed to be due to an FGF21-depentent reduction in membrane-associated diacylglycerols (DAG) and a corresponding decrease in proteinkinase Cθ (PKCθ) activity. Similar results were obtained with human skeletal muscle myotubes exposed to palmitate [37]. In accordance with such a mechanism, the NASH-diet resulted in an increase skeletal muscle inhibitory serine phosphorylation of IRS1 in the current study (Appendix A) that was attenuated by the training intervention. However, a direct impact of physical exercise on skeletal muscle IRS serine phosphorylation cannot be excluded. In addition, FGF21 has been shown to increase basal and insulin-induced glucose utilization in human myotubes partially by directly inducing Glut1 [23]. While Glut1 was not induced in skeletal muscle in the current study, another potential FGF21 target gene, UCP3 [24], was significantly increased in soleus muscle in the current study (Figure 7A). This in accordance with the observed increase in mitochondrial number (Figure 7B) might have contributed to reduced oxidative stress in skeletal muscle and thereby an improvement of insulin sensitivity in skeletal muscle.

## 5. Conclusions

While a seven-week endurance exercise training intervention did not improve diet-induced hepatic triglyceride accumulation and NAS it significantly reduced the diet-induced hepatic cholesterol overload and the ensuing severe oxidative stress. Assuming that oxidative stress is a crucial factor in further disease progression, this implies that continued training intervention might slow future disease progression despite the apparently stronger inflammatory response at the current time point of sampling. In addition, exercise enhanced hepatic FGF21 expression which might have contributed to improve systemic glucose tolerance without a training-dependent reduction of diet-induced weight gain.

## Figures and Tables

**Figure 1 nutrients-11-02709-f001:**
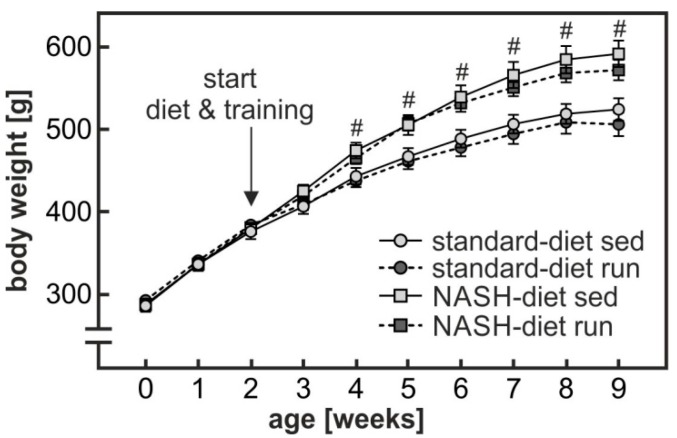
NASH-diet-dependent increase in body weight gain. After two weeks of adaptation (see Appendix A) rats received either a standard chow diet (standard-diet) or a high fat, high cholesterol, high fructose NASH-inducing diet (NASH-diet) for 7 weeks. Both diet groups were subdivided into a sedentary group (sed) and a group subjected to treadmill endurance exercise (run) (see methods section (paragraph 1) and Appendix A). Body weight was determined weekly. Values are means ± SEM. Statistics: Multiple Student’s *t*-test for unpaired samples; #: sedentary or exercised NASH-diet groups versus sedentary or exercised standard-diet groups, *p* < 0.05.

**Figure 2 nutrients-11-02709-f002:**
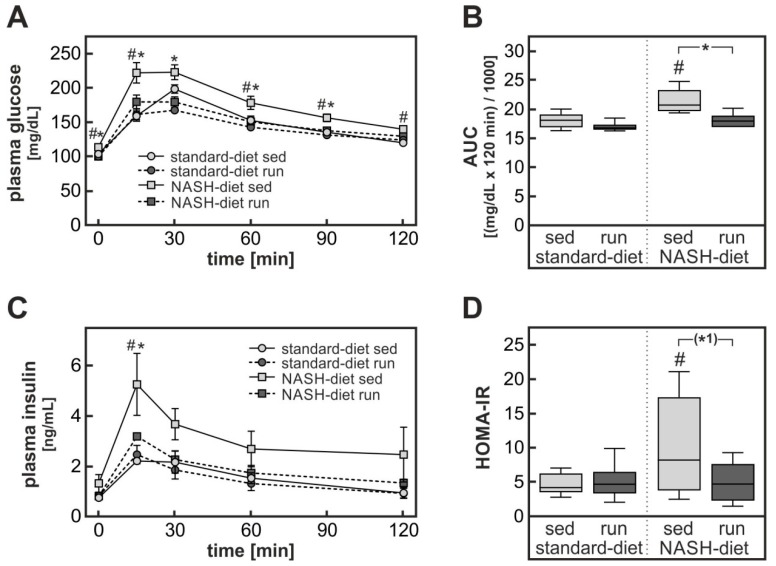
NASH-diet-dependent impairment of glucose tolerance and insulin sensitivity. Rats were subjected to the treatment groups described in the legend to Figure 1. One week before the end of the intervention an i.p. glucose tolerance test was performed (see methods section). (**A**) Blood glucose levels were determined enzymatically. Values are means ± SEM. (**B**) The area under the blood glucose curves (AUC) for every individual animal in (**A**) was determined. Values are median (line), upper and lower quartile (box) and extremes (whiskers). (**C**) Plasma insulin levels during the i.p. glucose tolerance test were determined by ELISA. Values are means ± SEM. (**D**) Homeostatic model assessment of insulin resistance (HOMA-IR) was calculated. Values are median (line), upper and lower quartile (box) and extremes (whiskers). Statistics: (**A**,**C**) Multiple Student’s *t*-test for unpaired samples; (**B**,**D**) Two-way-ANOVA with Tukey’s post hoc test for multiple comparisons; #: significant versus sedentary standard-diet group, *: significant versus exercised NASH-diet group, * *p* < 0.05, (*1) *p* < 0.10 between compared groups.

**Figure 3 nutrients-11-02709-f003:**
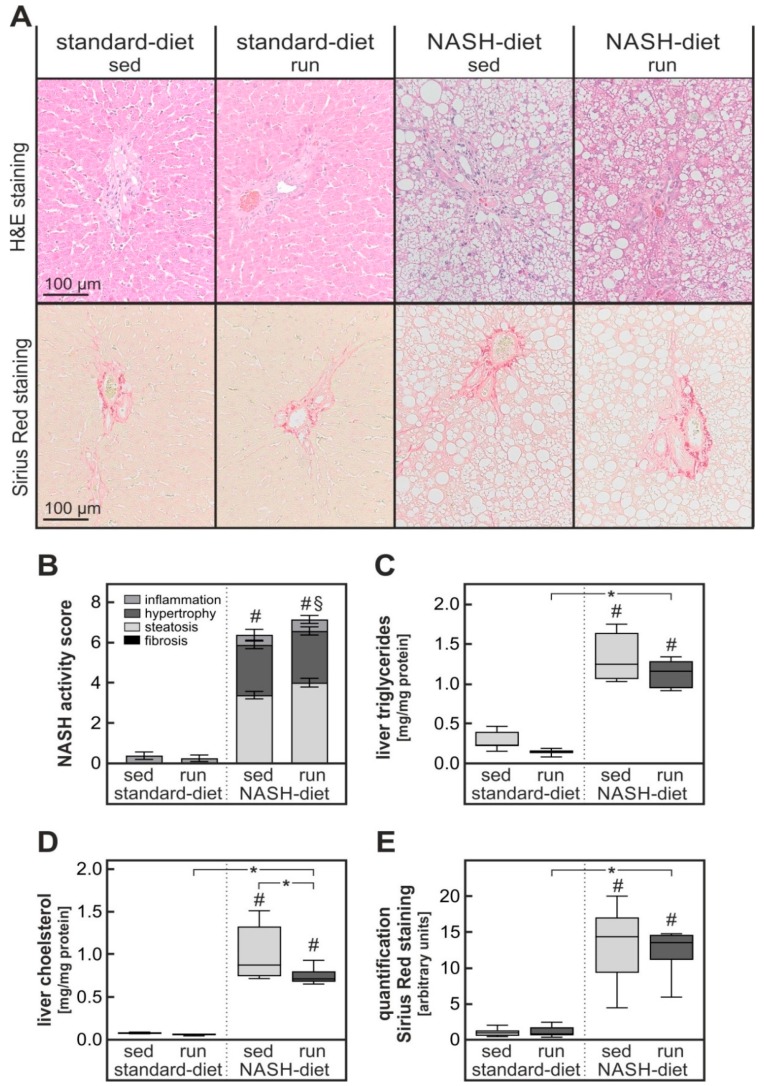
**Impact of diet and exercise on liver histology, NASH activity score, fibrosis, and hepatic lipid accumulation.** (**A**) Liver slices of the four intervention groups described in the legend to Figure 1 were stained with H & E or Sirius Red. Representative microphotographs, magnification 20×. (**B**) H&E-stained liver slices were scored according to a NASH activity score adapted for rodents [14] by a liver pathologist blinded to the treatment group. (**C**,**D**) Liver triglycerides (**C**) and cholesterol (**D**) were determined by enzymatic assays in liver homogenates. (**E**) The Sirius Red staining intensity was quantified densitometrically in parenchymal areas of the liver slices. Values are mean ± SEM (B) or median (line), upper and lower quartile (box) and extremes (whiskers) (**C**–**E**). Statistics: (**B**) Kruskal-Wallis test with Dunn’s post hoc test for multiple comparisons; (**C**–**E**) Two-way-ANOVA with Tukey’s post hoc test for multiple comparisons; #: significant versus sedentary standard-diet group, §: significant versus exercised standard-diet group, * *p* < 0.05 between compared groups.

**Figure 4 nutrients-11-02709-f004:**
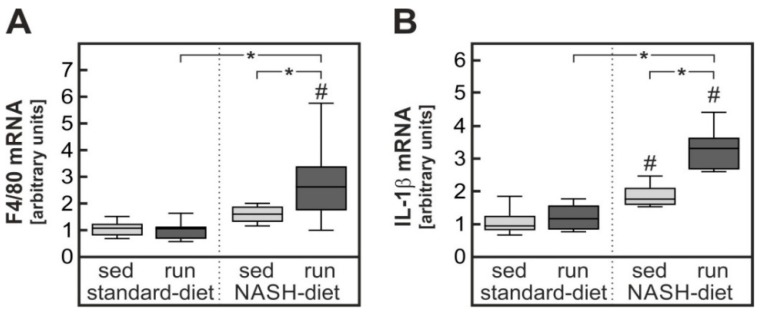
Enhanced inflammatory response in livers of NASH-diet fed trained rats. Rats were subjected to the different treatment groups described in the legend to Figure 1. The mRNA expression of F4/80 (gene name Adgre1, (**A**)) and Interleukin-1β (IL-1β, (**B**)) were quantified by RT-qPCR. Values are median (line), upper and lower quartile (box) and extremes (whiskers). Statistics: Two-way-ANOVA with Tukey’s post hoc test for multiple comparisons; #: significant versus sedentary standard-diet group, * *p* < 0.05 between compared groups.

**Figure 5 nutrients-11-02709-f005:**
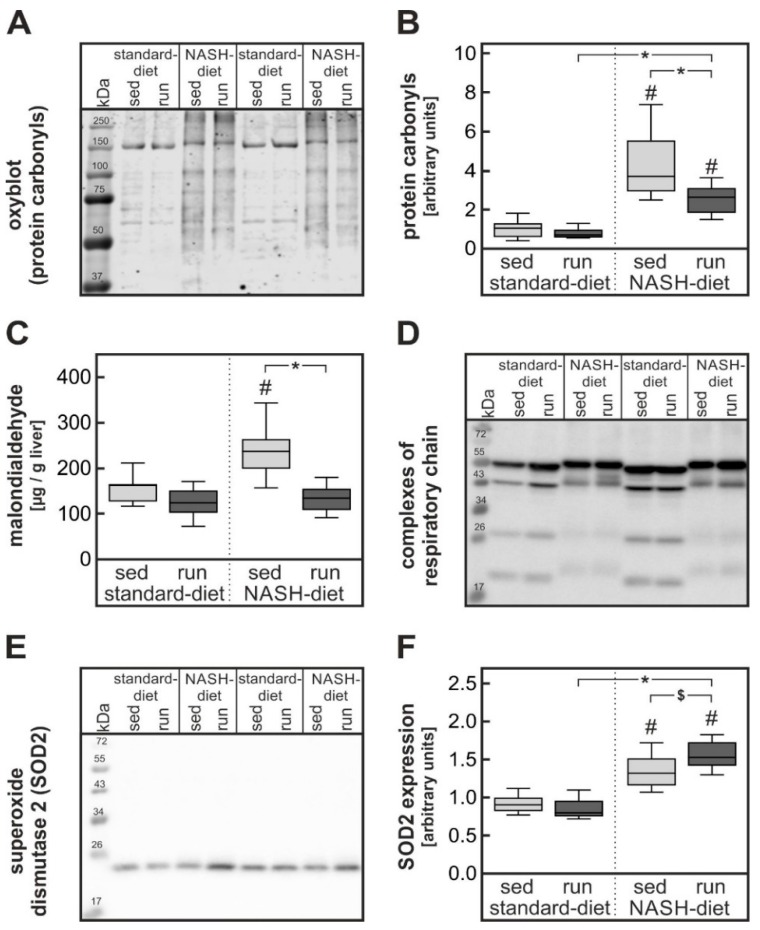
Exercise-dependent reduction of NASH-diet-induced hepatic oxidative stress and exercise-dependent increase of NASH-diet-induced hepatic anti-oxidative capacity. Parameters were determined in liver samples of rats of the different intervention groups described in the legend to Figure 1. (**A**,**B**) Protein carbonyls were detected by oxyblot as a measure of severe oxidative stress. A representative blot is shown (**A**). All blots, as well as Ponceau S staining as a loading control, are available as Appendix A. Blots of livers of all animals were subjected to densitometric quantification (**B**). (**C**) Malondialdehyde as a parameter for lipid peroxidation and mild oxidative stress was measured in liver homogenates by TBARS assay. (**D**) Hepatic expression of the complexes of the respiratory chain was determined by western blot. A representative blot is shown. All blots as well as the quantification are available as Appendix A. (**E**,**F**) Hepatic expression of superoxide dismutase 2 (SOD2) measured by western blot. A representative blot is shown (**E**). All blots are available as Appendix A. Blots of livers of all animals were subjected to densitometric quantification (SOD2 relative to FastGreen staining as a loading control) (**F**). Values are median (line), upper and lower quartile (box) and extremes (whiskers). Statistics: Two-way-ANOVA with Tukey’s post hoc test for multiple comparisons; #: significant versus sedentary standard-diet group, * *p* < 0.05 between compared groups; $: *p* = 0.070 between compared groups.

**Figure 6 nutrients-11-02709-f006:**
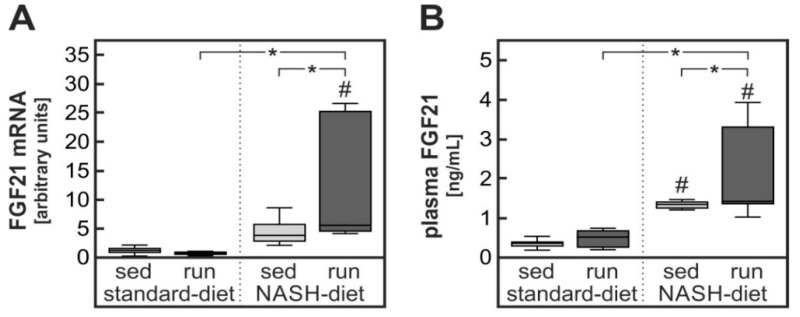
Diet- and exercise-dependent induction of hepatic FGF21 production. (**A**) Fibroblast growth factor 21 (FGF21) mRNA expression was quantified by RT-qPCR in liver samples from the different intervention groups described in the legend to Figure 1. (**B**) FGF21 plasma protein concentration was determined. Values are median (line), upper and lower quartile (box) and extremes (whiskers). Statistics: Two-way-ANOVA with Tukey’s post hoc test for multiple comparisons; #: significant versus sedentary standard-diet group, * *p* < 0.05 between compared groups.

**Figure 7 nutrients-11-02709-f007:**
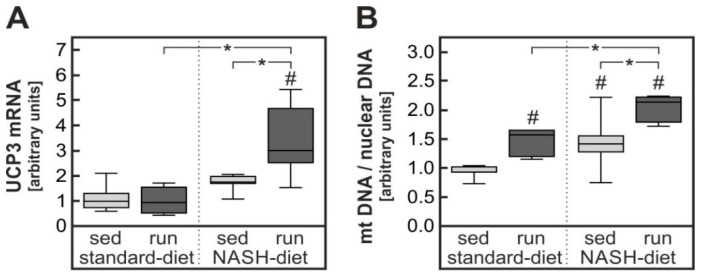
Diet- and exercise-dependent induction of UCP3 and mitochondrial DNA in skeletal muscle. mRNA, genomic DNA and mitochondrial DNA was extracted from soleus muscle of animals of the different intervention groups described in the legend to Figure 1. (**A**) mRNA of uncoupling protein 3 (UCP3) was quantified by RT-qPCR. (**B**) The copy number of a gene coded by mitochondrial DNA (NADH Dehydrogenase subunit 1) was compared to the copy number of a nuclear-coded gene (β-actin). Values are median (line), upper and lower quartile (box) and extremes (whiskers). Statistics: Two-way-ANOVA with Tukey’s post hoc test for multiple comparisons; #: significant versus sedentary standard-diet group, * *p* < 0.05 between compared groups.

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
