# Peer review of "Reduced Oxidative Stress and Enhanced FGF21 Formation in Livers of Endurance-Exercised Rats with Diet-Induced NASH"

_nutrients, 2019, doi:10.3390/nu11112709_

Round 1
Reviewer 1 Report
In the manuscript “Reduced oxidative stress and enhanced FGF21 formation in livers of endurance-exercised rats with diet-induced NASH”, the authors propose to study the impact of endurance exercise in a rat model of NASH. Interestingly, the authors concluded that exercise attenuated the diet-induced hepatic cholesterol overload oxidative damage though enhanced signs of inflammation. The work can be of interest but there are several drawbacks that hampered the initial enthusiasm. In addition, the experimental approach falls short for the claims. For instance, oxidative stress and redox homeostasis must be consolidated with further studies. The take home message is also very unclear. Further experiments should be performed to build a mechanistic-orientated hypothesis and discussion. I have some suggestions that may be useful.
Specific comments:
There are some minor language corrections needed. For instance: “…(NAFLD) that currently affects about one quarter of the world population [1] and more than every second type 2 diabetic patient”. The authors point is unclear. Exercise protocol is not well explained. Please include a figure and present it in a straightforward way. Results should be numerically described to highlight the most relevant differences and evaluate the possible significance. The study of oxidative stress-related effects is very incomplete as the authors mainly studied carbonylation. Lipid peroxidation, nitration as well as the antioxidant defences (Glutathione, SOD) should be determined to carefully check the redox balance and homeostasis before claiming that there is a reduced oxidative stress. Preferentially, ROS production should also be studied. To further support the mechanistic interpretation of the data, mitochondrial function should be studied and determined if it is affected and mediates the oxidative-stress related effects detected. Conclusion and “take home message” are no clear. The clinical/biological significance of the data should be highlighted. In addition, there are several study limitations that must be discussed. For instance, was the exercise protocol performed during the night, which is the active period for rats? If not, it should be discussed and cautions referred.Author Response
We thank the Reviewers for all their suggestions, effort and advice helping to improve our manuscript and making it clearer for the readers.
Reviewer 1:
In the manuscript “Reduced oxidative stress and enhanced FGF21 formation in livers of endurance-exercised rats with diet-induced NASH”, the authors propose to study the impact of endurance exercise in a rat model of NASH. Interestingly, the authors concluded that exercise attenuated the diet-induced hepatic cholesterol overload oxidative damage though enhanced signs of inflammation. The work can be of interest but there are several drawbacks that hampered the initial enthusiasm. In addition, the experimental approach falls short for the claims. For instance, oxidative stress and redox homeostasis must be consolidated with further studies. The take home message is also very unclear. Further experiments should be performed to build a mechanistic-orientated hypothesis and discussion. I have some suggestions that may be useful.
Specific comments:
There are some minor language corrections needed. For instance: “…(NAFLD) that currently affects about one quarter of the world population [1] and more than every second type 2 diabetic patient”. The authors point is unclear.
Authors’ comment:
We apologize for the misunderstanding. The particular sentence was rephrased. In addition, the entire manuscript was checked for language errors by a native speaker.
Exercise protocol is not well explained. Please include a figure and present it in a straightforward way.
Authors’ comment:
We thank the Reviewer for this remark. The exercise protocol is now explained and illustrated by a supplementary figure (Suppl. Figure S1). See also comment on day/night cycle.
Results should be numerically described to highlight the most relevant differences and evaluate the possible significance.
Authors’ comment:
As suggested by the Reviewer, the numerical differences are now mentioned in the text at those places where it appears appropriate. However, numerical differences are calculated from means. Since these values differ from the medians shown in the figures, this additional information in the text might be confusing. For the figures we chose to show medians, quartiles and extremes in the box and whisker plots to allow the reader to get a better idea of the inter-individual variations in the experiments and we still think that this is the most honest way to represent the data.
The study of oxidative stress-related effects is very incomplete as the authors mainly studied carbonylation. Lipid peroxidation, nitration as well as the antioxidant defenses (Glutathione, SOD) should be determined to carefully check the redox balance and homeostasis before claiming that there is a reduced oxidative stress. Preferentially, ROS production should also be studied.
Authors’ comment:
As suggested by the Reviewer, additional experiments to determine oxidative stress and anti-oxidative capacity have been performed. The hepatic concentrations of malondialdehyde as well as the expression of superoxide dismutase 2 and the activity of glutathione peroxidases are now incorporated into Figure 5C, E, F and in the text (page 8, line 266 ff. and page 9, line 275 ff.). Although we agree with the Reviewer that it would be useful to determine ROS production, this can no longer be done in post mortem samples.
To further support the mechanistic interpretation of the data, mitochondrial function should be studied and determined if it is affected and mediates the oxidative-stress related effects detected
Authors’ comment:
As with the ROS determination, mitochondrial function can only be tested in fresh tissue. This was not possible in the current experimental setup. As a surrogate for the mitochondrial function, we determined the expression of the complexes of the mitochondrial respiratory chain by western blot. These data are now included in Figure 5D and in the Suppl. Figure S3.
Conclusion and “take home message” are no clear. The clinical/biological significance of the data should be highlighted.
Authors’ comment:
According to the Reviewers suggestion, we rephrased our conclusion and included some biological significance. However, we think that one has to be extremely careful not to overstress the translational relevance of the current animal experiment. Therefore we refrained from pointing out a potential clinical relevance.
In addition, there are several study limitations that must be discussed. For instance, was the exercise protocol performed during the night, which is the active period for rats? If not, it should be discussed and cautions referred.
Authors’ comment:
We apologize that we did not correctly describe this aspect of the condition of the animal experiment. As now clearly stated in the new Suppl. Figure S1, animals were adapted to an inverted day/night cycle and trained in the dark phase (their activity phase).

Reviewer 2 Report
Excellent hypothesis testing, however, exercise regimen per se did not reduce any bodyweight in rats. i.e. the effect of the NASH diet was able to override effects of exercise in terms of adiposity. Was any body fat composition analysis performed? Rest of the metabolic panels look reasonable and explain the effects differentially on systemic glucose homeostasis vs. increased acute inflammatory markers in exercise+cholesterol group. Have the authors measured plasma cholesterol and TG levels, especially VLDL TG and secretion rates and fasting plasma NEFA levels, to be able to interpret indirect effects on insuslin sensivity reduced lipolysis. Most critically, have the authros tried to use just a high-fat high carbohydrate diet and subjected those to exercise regimen? this will be a true control since cholesterol seems to drive acute liver injury in current study and is hard to separate aspects related to NAFLD from NASH. I understand that the focus is to develop a NASH model but it's impossible to physiologically sperate the effects in the current study, lastly what is the effect on plasma transaminases on these various conditions and MUST be included.
Author Response
We thank the Reviewers for all their suggestions, effort and advice helping to improve our manuscript and making it clearer for the readers.
Reviewer 2:
Excellent hypothesis testing, however, exercise regimen per se did not reduce any bodyweight in rats. i.e. the effect of the NASH diet was able to override effects of exercise in terms of adiposity. Was any body fat composition analysis performed?
Authors’ comment:
We agree with the Reviewer, but unfortunately, body composition could not be tested in the current study due to technical limitations (i. e. the NMR-equipment for the only currently validated protocol for the determination of the body composition was too small to fit the older rats.)
Rest of the metabolic panels look reasonable and explain the effects differentially on systemic glucose homeostasis vs. increased acute inflammatory markers in exercise+cholesterol group. Have the authors measured plasma cholesterol and TG levels, especially VLDL TG and secretion rates and fasting plasma NEFA levels, to be able to interpret indirect effects on insulin sensitivity reduced lipolysis.
Authors’ comment:
Plasma cholesterol and triglyceride levels were already shown in Suppl. Table S2. As suggested by the Reviewer, the information about free fatty acid levels was now also included into this table (Suppl. Table S2). Although it would certainly be interesting to measure VLDL-secretion rat, this is not possible in the current experimental setting with reasonable effort, since this would require feeding, training and sacrificing additional animals only for this purpose. This is, because determination of the VLDL-secretion rate requires a pharmacological inhibition of the lipoprotein lipase, a deadly intervention for the animal, which cannot be used for the determination of any other parameter.
Most critically, have the authors tried to use just a high-fat high carbohydrate diet and subjected those to exercise regimen? This will be a true control since cholesterol seems to drive acute liver injury in current study and is hard to separate aspects related to NAFLD from NASH. I understand that the focus is to develop a NASH model but it's impossible to physiologically separate the effects in the current study.
Authors’ comment:
We agree with the Reviewer that a detailed analysis of the impact of different components of the diet on NASH development is extremely important. However, this was subject of previous publications of our group. In these studies, different diets with and without cholesterol were directly compared. The key finding was that only the combination of cholesterol and ω6-PUFA was able to trigger severe oxidative stress and NASH development (Henkel et al., Mol Med 2017 (PMID: 28332698) and Henkel et al., nutrients 2018 (PMID: 30231595)). The focus of the current study was to investigate the effect of training on the diet-induced steatohepatitis using the diet composition that had been shown to be most effective in inducing NASH in the previous studies.
lastly what is the effect on plasma transaminases on these various conditions and MUST be included.
Authors’ comment:
We agree with the reviewer. ASAT was already shown in the original manuscript in Suppl. Table S2, we now also included the information of ALAT in Suppl. Table S2.

Round 2
Reviewer 2 Report
The author's revision looks good for me.